# Distortion Detection of Lithographic Projection Lenses Based on Wavefront Measurement

**Tian Li** [1,2], **Jian Wang** [1,3], **Shaolin Zhou** [2], **Haiyang Quan** [1,*], **Lei Chen** [1], **Junbo Liu** [1], **Jing Du** [1], **Xianchang Zhu** [1,3] and **Song Hu** [1,3]

1    Institute of Optics and Electronics, Chinese Academy of Sciences, Chengdu 610209, China
2    School of Microelectronics, South China University of Technology, Guangzhou 510641, China
3    University of Chinese Academy of Sciences, Beijing 100049, China
*    Correspondence: haiyangquan@ioe.ac.cn

**Abstract:** As with the decreasing feature size prompted by Moore's law and the continuous technological advancements in the semiconductor industry, the distortion of the projection lens is an important factor that affects the overlay. In this paper, we propose a wavefront-measurement-based method to detect the projection lens distortion in the lithographic system. By normalizing the coordinates of the Shack–Hartmann system with the image displacements represented in the $Z_2$ and $Z_3$ terms of Zernike coefficients, the offsets between the actual image points and the ideal image points can be determined. By offset collection at an array of $7 \times 7$ field points to establish an overdetermined system of equations, the proposed method can simultaneously detect the distortions of translation, magnification, rotation, decentering distortion, thin prism distortion, and third-order radial distortion. This distortion measurement method is highly flexible for distortion measurement with portable and compactly integrated sensors, enabling the real-time and cost-efficient measurement of wave aberration and distortion. For proof-of-concept experiments, a projection lens with a numerical aperture (NA) of 0.58 for i-line (365 nm) is used for experimental testing. The results reveal that the repeatability accuracy of distortion detection is 51 nm and the 72 h long-term reproducibility is 143 nm.

**Keywords:** distortion; lithography; projection lens; wavefront measurement

## 1. Introduction

Over the past few decades, the complexity and density of integrated circuits (IC) have been increasing with Moore's law and thus the on-chip feature size keeps shrinking. To ensure the accurate transfer of the mask pattern onto the wafer with tolerable errors, the image distortion detection of photolithographic projection lenses is imperative. The distortion represents the deviation of the actual position away from the ideal position at different points in the image plane. Such image distortions are mainly caused by stage errors and the lens magnification and aberration errors of the lithographic system.

Current methods of distortion detection for typical lithographic projection lenses include exposure measurement, aerial image measurement, and wavefront measurement. First, Litel proposed a self-referencing method to measure the distortion of the projection lens in 2003, in which the Box-in-Box scheme was used in exposure measurement to reduce the repeatability accuracy to less than 1 nm [1]. Moreover, commercial lithography companies GCA, ASML, and Canon have all conducted in-depth research on this method [2–4]. The exposure method is the most widely used and mature route that has high accuracy but low cost-efficiency depending on the time-consuming overlay process, including iterative ultraviolet (UV) light exposure, photoresist development, and etching.

Secondly, aerial image measurement aims to detect the projected images that are "floating in air" but cannot be visually observed directly. The aerial image distortion can be retrieved by scanning the stage across the image plane using one pair of conjugated marks

on the stage and the mask. In such a manner, the light intensity field that represents the convolution of two conjugated marks can be recorded to calculate the offset distributions through optical-to-electrical signal conversion and scanning data fitting to obtain the final distortion parameters. Both Nikon and ASML have adopted this technique in different manners, in which Nikon performs slit scans along the $x$ and $y$ axes to obtain an aerial image [5], whereas ASML acquires an aerial image using a transmission image sensor (TIS) for the lateral scanning of alignment marks on the wafer plane [6]. The method of aerial image measurement can be performed online and process-free with high stability, and this not only makes it difficult to process data from many sampling points, but also means that the detection speed is dependent on the costly high-speed stage movement.

Finally, the wavefront measurement method essentially encodes the image distortions into wavefront deformations after the mask and projection lens. The offset of the actual image point with regard to the ideal image point can be decoded by wavefront measurement to ultimately fit the as-required distortion parameters. As typical examples, Sandia National Laboratory developed a point diffractometer-based method for distortion detection in 2001 [7], and ASML proposed a shear interference-based method for distortion detection in 2003 [8] and cooperated with Zeiss to study the limitations of the third-order distortion, and they proposed an improved method to predict the etching error and control the high-order distortion [9,10]. The wavefront measurement method detects both distortions and aberrations simultaneously, in a low cost and exposure-free process that also depends on the positioning and measurement accuracy of the moving stage.

The Shack–Hartmann wavefront sensor, commonly used for aberration detection [11–13] and alignment [14,15] in adaptive optics, is readily adopted to retrieve the wavefront deformation by sequentially scanning an array of marks on the mask through the projection lens. Finally, the lateral offset between actual positions and ideal positions can be calculated by decoding the wavefront aberration of the projection lens. The cost of wavefront detection is lower than that of an interferometer.

In this paper, we propose a scheme for projection lens distortion detection via the integration of a Shack–Hartmann wavefront sensor. To characterize the distortion components, including translation, rotation, magnification, decentering distortion, thin prism distortion, and third-order radial distortion, the first two Zernike coefficients are detected by the integrated Shack–Hartmann sensor and superimposed with the coordinates of the moving stage. For experimental testing at a set of 49 field points in the image plane, the offsets between the actual image points and the ideal image points were detected by establishing a system of overdetermined equations. The experiments were conducted with a 365 nm projection lens with a numerical aperture (NA) of 0.58. The distortion of $7 \times 7$ field points in the ideal position of the image plane and the actual position are scanned by the stage bearing the Shack–Hartmann sensor; the distortion detection model can give the distortion coefficients of the lens, and we adjust the distortion of the experimental lens' $x$-axis and $y$-axis from 2634.61 nm and 3005.19 nm to 177.12 nm and 218.85 nm through dynamic mirror adjustment. The repeatability accuracy of distortion detection was measured to be 51 nm, with the short-term and long-term reproducibility of 142 nm and 143 nm. Compared with the above measurement method, our proposed method and model not only provide a solution for distortion but also for decentering and thin prism distortion parameters, which can be used to guide the optical design to make further corrections to the projection lens system. These two parameters can help us to evaluate the performance of the lens inside the projection lens and to adjust the lens inside the projection lens so that the distortion can be reduced. This method is flexible, the sensor is small and portable, it can detect wave aberration and distortion in real time, and it is inexpensive.

The remainder of this paper is organized as follows: the Section 2 defines each component of distortion with adequate modeling; the 'Method' section describes the principles and experimental system for distortion detection by integrating the Shack–Hartmann sensor; the 'Empirical Study' section introduces the mask design and sampling strategy with the involved parameters of specifications. Finally, the 'Results and Discussion' section

describes and analyzes the calculated experimental results, and the 'Conclusions' section highlights the main contributions and potential for future work.

## 2. Model

Distortion is a key indicator of the quality of a projection lens system in photolithography. A variety of factors influence distortion, including the illumination system, mask design, projection lens, working environment, stage, and measurement tools and methods.

The notations and coefficients used in this study are listed in Table 1.

**Table 1.** Definitions of some notations and coefficients used in this study.

| Notations and Coefficients | Definitions |
| --- | --- |
| $dx, dy$ | distortion offsets along the $x$-axis and $y$-axis of the image plane |
| $x, y$ | coordinates of the image plane |
| $T_x, T_y$ | translations along the $x$-direction and $y$-direction |
| $\theta_x, \theta_y$ | rotations with regard to the $x$-axis and $y$-axis |
| $M_x, M_y$ | magnifications along $x$-axis and $y$-axis |
| $B_x, B_y$ | bow coefficients along $x$-axis and $y$-axis |
| $T_{xx}, T_{yx}$ | trapezoid distortion along the $x$-axis |
| $T_{yy}, T_{xy}$ | trapezoid distortion along the $y$-axis |
| $W_x, W_y$ | wedge distortion along $x$-axis and $y$-axis |
| $p_{11}$ | the first-order decentering distortion coefficient |
| $p_{12}$ | the second-order decentering distortion coefficient |
| $q_{11}, q_{12}$ | thin prism distortion coefficient along the $x$-axis and $y$-axis |
| $D_{3x}, D_{3y}$ | the third-order radial distortion coefficient along $x$-axis and $y$-axis |
| $r$ | radial deviation of origin point |
| $r_x, r_y$ | residuals along $x$-axis and $y$-axis |

Intrinsically, distortion can be modeled by a variety of parameters, including translation, rotation, and expansion [16], i.e.,

$$dx = T_x - \theta_x y + M_x x$$
$$dy = T_y + \theta_y x + M_y y \tag{1}$$

Specifically, for the stepper system [17], more parameters, including translation, rotation, magnification, lens trapezoidal distortion, and third-order radial distortion, are used to model the distortions of the Zeiss' lens, namely,

$$dx = T_x - \theta_x y + M_x x + T_{xx} x^2 + T_{yx} xy + D_{3x} xr^2$$
$$dy = T_y + \theta_y x + M_y y + T_{xy} xy + T_{yy} y^2 + D_{3y} yr^2 \tag{2}$$

Further, Perloff et al. proposed a model to perform a superposed error analysis on a 1:1 projection aligner and a 10:1 wafer stepper [18], considering not only translation, rotation, and magnification, but also lens distortion and the effect of stage parameters, as follows:

$$dx = T_x - \theta_x y + M_x x + B_x y^2$$
$$dy = T_y + \theta_y x + M_y y + B_y x^2 \tag{3}$$

Then, based on the above model, and the assistance of grating alignment technology [19–21] through a lens, [22] extended the radial distortion term of the model to the fifth order and took into account the effect of both inter-field and intra-field errors as follows:

$$
\begin{aligned}
dx &= T_x - \theta_x y + M_x x + T_{xx} x^2 + T_{yx} xy + W_x y^2 + D_{3x} x r^2 + D_{5x} x r^4 + r_x \\
dy &= T_y + \theta_y x + M_y y + T_{xy} xy + T_{yy} y^2 + W_y x^2 + D_{3y} y r^2 + D_{5y} y r^4 + r_y
\end{aligned}
\tag{4}
$$

In practice, [23] considered the magnification and orthogonality on the *x*-axis and *y*-axis during the stepping process. Ref. [24] studied the relationship between sampling and global alignment repeatability, and studied the data through different sampling methods to find the importance of a data sample with central symmetry and good sampling coverage. A two-stage sampling strategy is proposed to effectively estimate and compensate for the distortion by using a small number of samples [25]. Further, Chien et al. also proposed a UNISON decision analysis procedure and developed a distortion model for step and scan distortion, in which an optimal sampling strategy was involved for measuring and distortion compensation [26]. To fill the gap between the existing theoretical model and the actual data, the sampling strategy based on the empirical data of the wafer factory was discussed [27]. The weighted least square method is used to describe the distortion detection more accurately and compared with the least square method under the same conditions [28]. A distortion solution model with 20 parameters is proposed to further reduce the calculation residual [29].

Overall, the parameters and sampling strategies for various schemes of distortion detection have been intensively studied in the above models, but little attention has been paid to decentering distortion and thin prism distortion in projection systems, and the physical meaning of some parameters is not clear.

### 2.1. Decentering Distortion

The decentering error induced in the process of optical mounting is not axisymmetric and is primarily responsible for decentering distortion. Both decentering and thin prism distortion cause radial and tangential distortions. The decentering distortion is expressed as follows [30]:

$$
\begin{aligned}
\delta_{\rho d} &= 3\left(j_1 \rho^2 + j_2 \rho^4 + \cdots + j_n \rho^{2n}\right) \sin(\varphi - \varphi_0) \\
\delta_{td} &= \left(j_1 \rho^2 + j_2 \rho^4 + \cdots + j_n \rho^{2n}\right) \cos(\varphi - \varphi_0)
\end{aligned}
\tag{5}
$$

where $\delta_{\rho d}$, $\delta_{td}$ denote the amount of decentering distortion along the radial and tangential directions, $j_1, j_2 \ldots j_n$ denote the decentering distortion coefficients, $\rho$ is the polar radius, $\varphi$ is the polar angle in the polar coordinate system, and $\varphi_0$ is the polar angle for the maximum tangential distortion. In the Cartesian coordinate system, the decentering distortion is expressed as

$$
\begin{aligned}
\delta_{xd} &= p_{11} xy + p_{12}\left(3x^2 + y^2\right) + \cdots + \left(p_{n1} xy + p_{n2}\left(3x^2 + y^2\right)\right) \rho^{2n-2} \\
\delta_{yd} &= p_{11}\left(x^2 + 3y^2\right) + p_{12} xy + \cdots + \left(p_{n1}\left(x^2 + 3y^2\right) + p_{n2} xy\right) \rho^{2n-2}
\end{aligned}
\tag{6}
$$

where $\delta_{xd}$, $\delta_{yd}$ denote the amount of decentering distortion along the *x*- and *y*-directions, and $p_{11}, p_{12} \ldots p_{n1}, p_{n2}$ denote the decentering distortion coefficients.

### 2.2. Thin Prism Distortion

The thin prism distortion is caused by the tilt error in the mounting process of the optical element or detector, and its actual effect is similar to that of a thin prism. The thin prism distortion is expressed as follows [31]:

$$
\begin{aligned}
\delta_{\rho d} &= \left(i_1 \rho^2 + i_2 \rho^4 + \cdots + i_n \rho^{2n}\right) \sin(\varphi - \varphi_0) \\
\delta_{td} &= \left(i_1 \rho^2 + i_2 \rho^4 + \cdots + i_n \rho^{2n}\right) \cos(\varphi - \varphi_0)
\end{aligned}
\tag{7}
$$

where $\delta_{\rho d}$, $\delta_{td}$ denote the amount of thin prism distortion along the radial and tangential directions, $i_1, i_2 \ldots i_n$ are the thin prism distortion coefficients, $\varphi$ is the polar angle in the

polar coordinate system, and $\varphi_0$ is the polar angle for the maximum tangential distortion. In the Cartesian coordinate system, the thin prism distortion is expressed as

$$\begin{aligned} \delta_{xp} &= q_{11}\rho^2 + q_{21}\rho^4 + \cdots + q_{n1}\rho^{2n} \\ \delta_{yp} &= q_{12}\rho^2 + q_{22}\rho^4 + \cdots + q_{n2}\rho^{2n} \end{aligned} \tag{8}$$

where $\delta_{xp}$, $\delta_{yp}$ denote the amount of thin prism distortion along the $x$- and $y$-directions, and $q_{12}, q_{21}, q_{22} \ldots q_{n1}, q_{n2}$ are the thin prism distortion coefficients.

### 2.3. The Proposed Distortion Model

Based on all the aforementioned models, we intend to propose a distortion detection model to incorporate all components of translation, rotation, magnification, decentering distortion, thin prism distortion, and third-order radial distortion together, as shown in Figure 1. For light propagation through the projection lens, the actual image points deviate from their ideal positions in the image plane along the $x$- and $y$-directions due to all error components of translation, rotation, magnification, decentering distortion, thin prism distortion, and third-order radial distortion. Further, systematic distortion errors due to mask manufacturing errors and the wafer process also cause image point deviations.

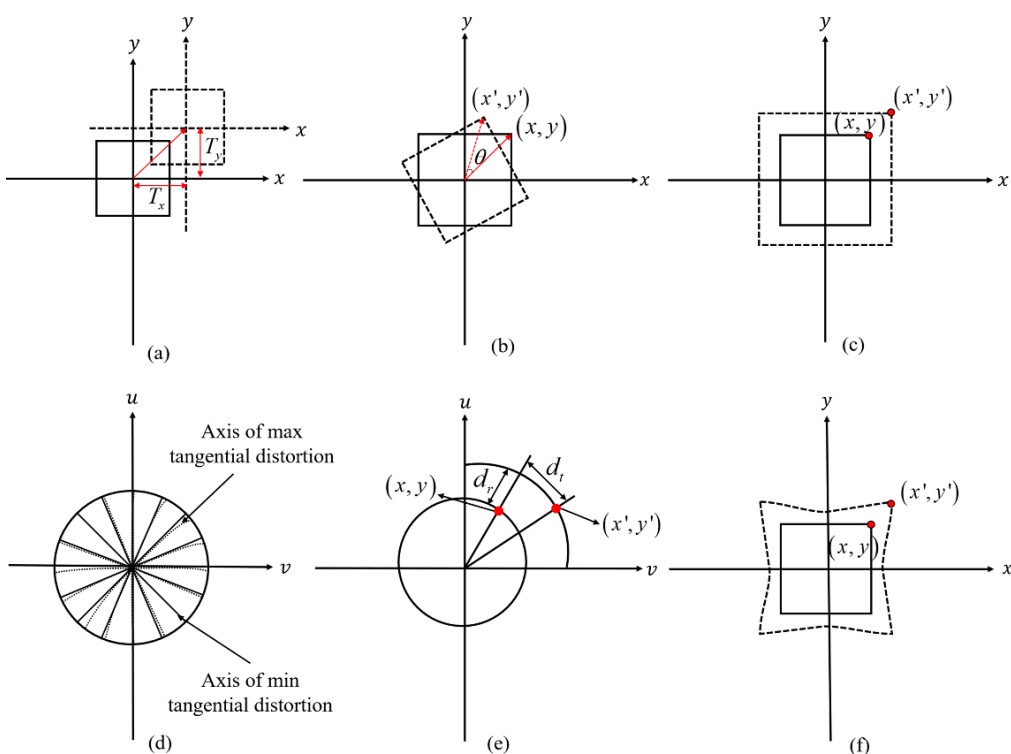

**Figure 1.** Different error components that cause distortions. (**a**) Translation. (**b**) Rotation. (**c**) Magnification. (**d**) Decentering distortion. (**e**) Thin prism distortion. (**f**) Third-order radial distortion.

Therefore, by integrating all offsets between the ideal image points and actual image positions caused by different error components, the final overall distortion can be modeled as

$$\begin{aligned} dx &= T_x - \theta_x y + M_x x + p_{11}xy + p_{12}(3x^2 + y^2) + q_{11}r^2 + D_{3x}xr^2 + r_x \\ dy &= T_y + \theta_y x + M_y y + p_{11}(x^2 + 3y^2) + p_{12}xy + q_{12}r^2 + D_{3y}yr^2 + r_y \end{aligned} \tag{9}$$

Further, the components that cause various types of distortion errors are also depicted in vector displacement maps, as shown in Figure 2.

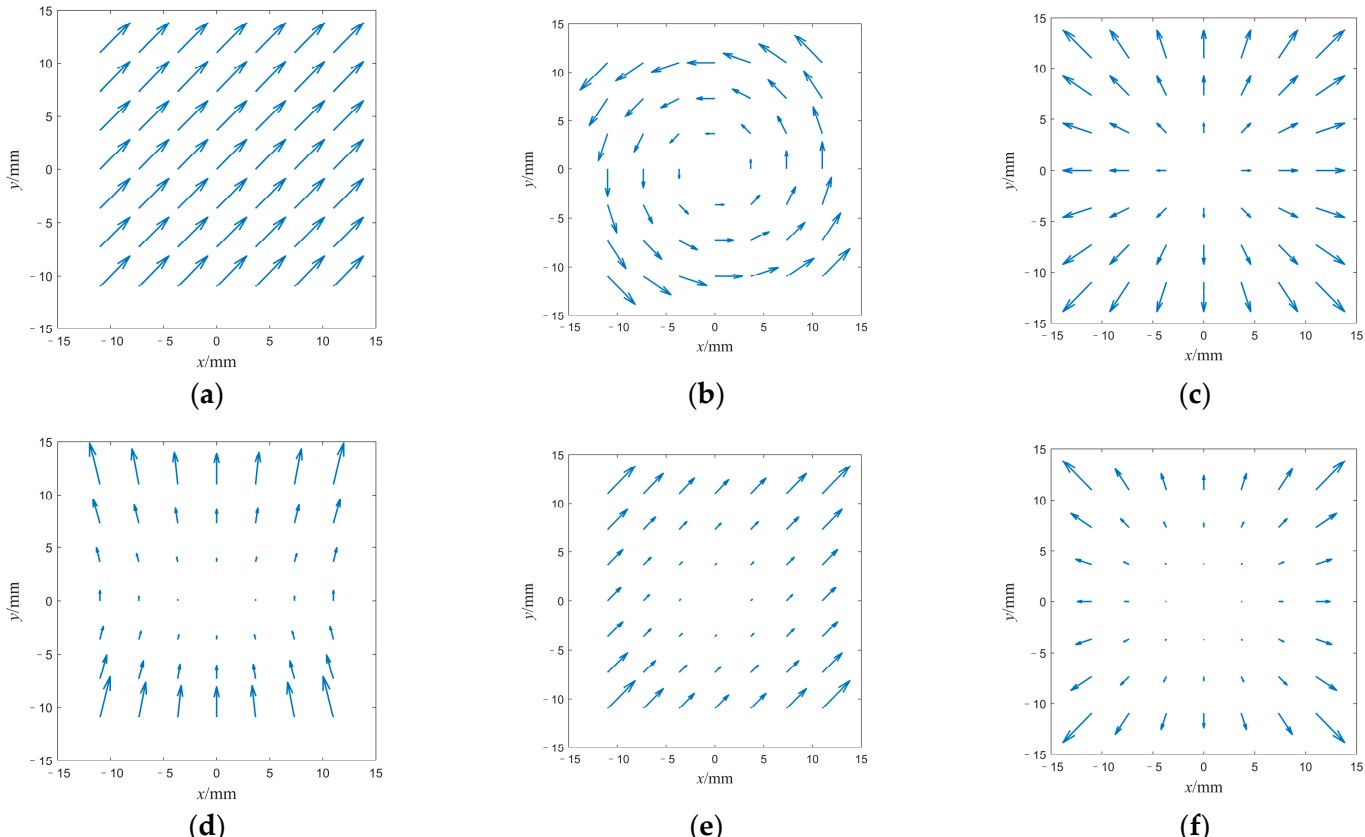

**Figure 2.** Vector map of different distortions. (**a**) Translation distortion. (**b**) Rotation distortion. (**c**) Magnification distortion. (**d**) Decentering distortion. (**e**) Thin prism distortion. (**f**) Third-order radial distortion.

## 3. Methods

To detect the overall distortions, the Shack–Hartmann wavefront detector, composed of a micro lens array and a charge-coupled device (CCD), is adopted to measure the aberrated wavefront first in this work. As shown in Figure 3, the array of micro lenses with equal focal length act as sub-apertures with the same size to divide the wavefront to be detected into an array of discrete sub-wavefronts. Each sub-aperture or sub-wavefront corresponds to a pixel of CCD located at the focal plane, with the micro lens array and the CCD image plane fixed in position. After light propagation through the micro lens array, the incident wavefront is spatially divided and collected by such sub-apertures to form an array of dots in its focal plane, as shown in Figure 3. By scanning the imaging plane with the Shack–Hartmann sensor, the actual image point positions of multiple field points are found, and the distortion of all the actual image points in the *x*- and *y*-directions in the coordinate system of the Shack–Hartmann sensor is compared with the coordinates of the stage to obtain the distortion of all the actual image points in the *x*- and *y*-directions from the ideal image point positions. The distortion of all the field points in the *x*- and *y*-directions are substituted into Equation (4) to obtain the distortion parameters.

When the ideal plane wave is incident on the micro lens array, an equally spaced array of spots appears on the CCD surface, with each spot located in the center of each pixel. When an actual wavefront with aberration is incident on the micro lens array, the center of mass or the spot of each sub-pixel array deviates from the centroid. By measuring the deviation of each spot for each sub-wavefront, the coordinates of the center of mass and the average slope of each sub-wavefront for each sub-aperture can be found and the overall wavefront can be reconstructed. The reconstructed wavefront can be represented by Zernike polynomials as

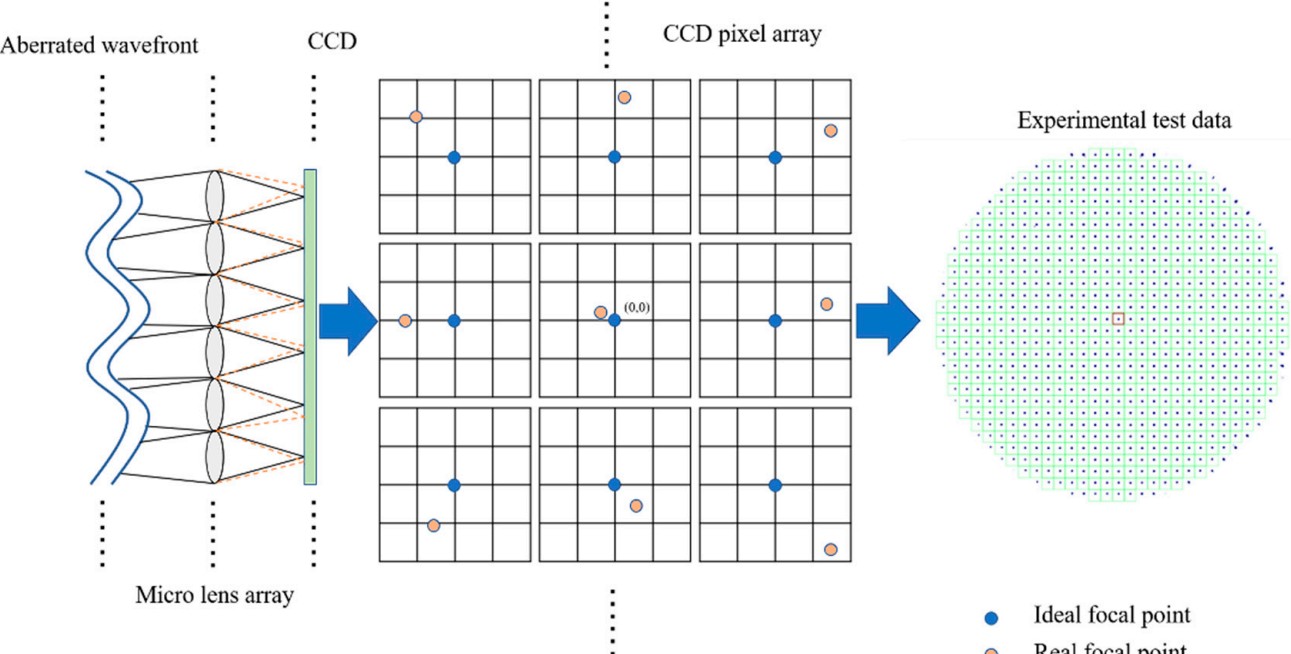

**Figure 3.** The scheme of the Shack–Hartmann wavefront sensor for distortion measurement.

$$
\begin{bmatrix}
G_x(1) \\
G_y(1) \\
G_x(2) \\
G_y(2) \\
\cdots \\
G_x(m) \\
G_y(m)
\end{bmatrix}
=
\begin{bmatrix}
Z_{x1}(1) & Z_{x2}(1) & \ldots & Z_{xn}(1) \\
Z_{y1}(1) & Z_{y2}(1) & \ldots & Z_{yn}(1) \\
Z_{x1}(2) & Z_{x2}(2) & \ldots & Z_{xn}(2) \\
Z_{y1}(2) & Z_{y2}(2) & \ldots & Z_{yn}(2) \\
\cdots & \cdots & \cdots & \cdots \\
Z_{x1}(m) & Z_{x2}(m) & \ldots & Z_{xn}(m) \\
Z_{y1}(m) & Z_{y2}(m) & \ldots & Z_{yn}(m)
\end{bmatrix}
\begin{bmatrix}
a_1 \\
a_2 \\
\cdots \\
a_n
\end{bmatrix}
\tag{10}
$$

where $G_{x_m}$, $G_{y_m}$ are the $x$ and $y$ slopes of the spot for the $m$th sub-aperture, $Z_{x_n}$, $Z_{y_n}$ are the average slopes of the aberrations represented by the $n$th Zernike polynomial in the $x$- and $y$-directions at this sub-aperture, $a_n$ is the coefficient of the $n$th Zernike polynomial, $m$ is the number of sub-apertures, and $n$ is the order of the Zernike polynomial.

Thereafter, deviations of each spot along the $x$- and $y$-directions can be calculated from the 2nd and 3rd terms of the Zernike polynomial, i.e., the coefficients of the $Z_2$ and $Z_3$ terms that can be obtained from Equation (10),

$$
\begin{aligned}
\Delta s_{x_i} &= f \times \frac{a_2}{R} \\
\Delta s_{y_i} &= f \times \frac{a_3}{R}
\end{aligned}
\tag{11}
$$

where $\Delta s_{x_i}$, $\Delta s_{y_i}$ are the offset of the $i$th field point from its ideal focal point along the $x$- and $y$-directions, $f$ is the focal length of the collimating lens of the Shack–Hartmann sensor, and $R$ is the radius of the unit circle detected by the CCD.

The final offset between the actual image position and the ideal image position is obtained by calibrating the $x$ and $y$ offset measured by the Shack–Hartmann sensor to the global coordinates determined by the interferometer, i.e.,

$$
\begin{aligned}
\Delta x_i &= X_i - \Delta s_{x_i} \\
\Delta y_i &= Y_i - \Delta s_{y_i}
\end{aligned}
\tag{12}
$$

where $X_i$ and $Y_i$ denote the ideal image positions of the $i$th field point in the image plane. In total, the offsets of 49 points are calculated and introduced to establish the model of overdetermined equations. Finally, the distortion parameters are fitted by using the least squares method.

For our experiment, as shown in Figure 4, a Shack–Hartmann sensor integrated system is established for the distortion detection of a typical projection lens. A 365 nm wavelength LED light source is used for illumination. As mentioned above, the $Z_2$ and $Z_3$ coefficients are obtained and converted into offsets along the $x$- and $y$-directions, and finally normalized to the interferometer's global coordinates. The data of an array of 49 field points or positions were obtained by point-by-point Shack–Hartmann scanning to build the final computational model, which can be solved by using a least square fitting procedure.

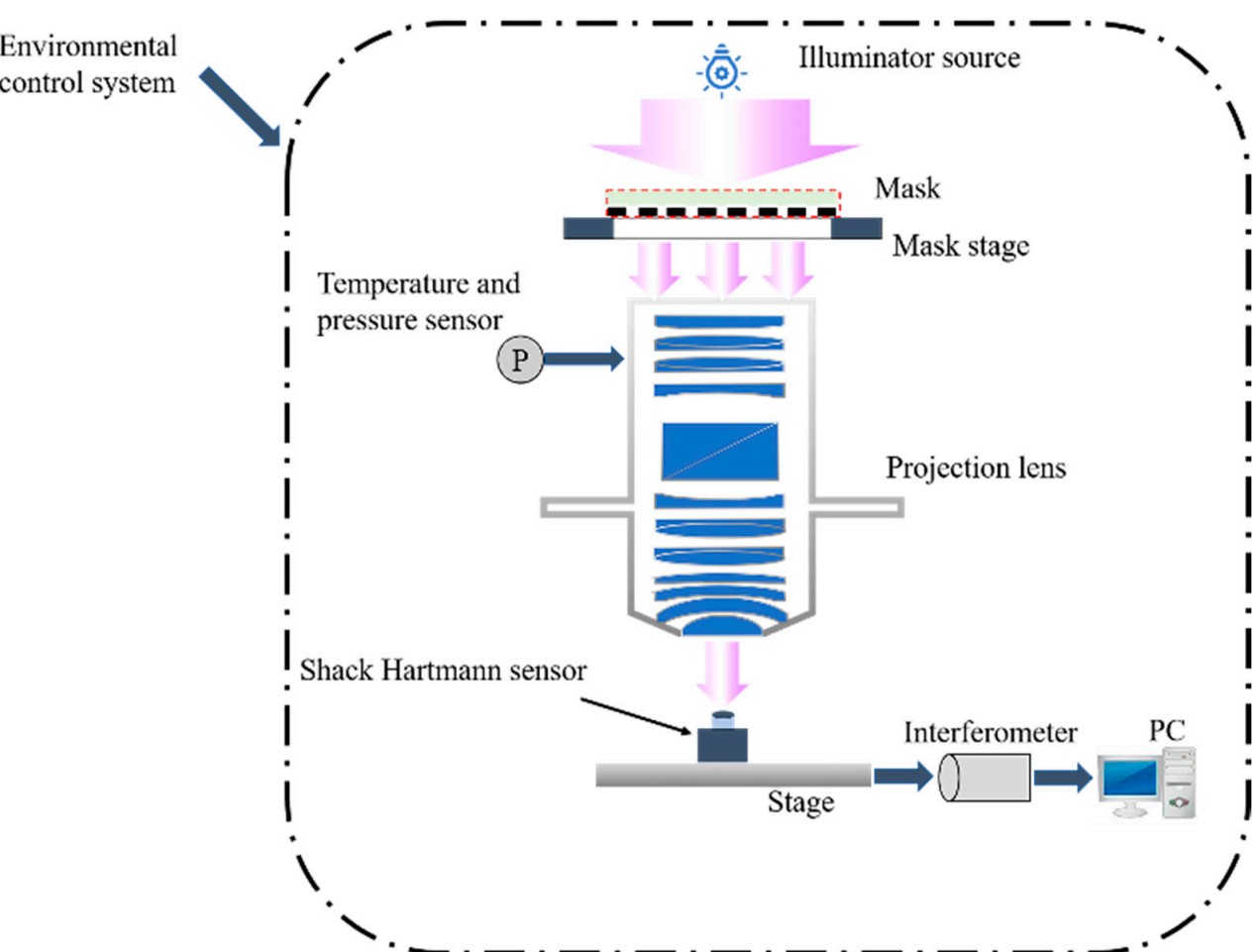

**Figure 4.** The diagram of our Shack–Hartmann sensor integrated system for distortion detection.

## 4. Empirical Study

For empirical study using the distortion detection method, the number of sampling points determines the accuracy of the calculated parameters, i.e., more sampling points lead to higher computational accuracy but a simultaneously larger cost concerning the storage space and sampling time. The mask size chosen for the experiment is 6 inches and it is divided into 11 × 11 grids, as shown in Figure 5. Each grid's size is 10 mm × 10 mm and the yellow area with a size of 4.5 μm × 4.5 μm is located in the center for incident light transmission. For a reasonable trade-off between the experimental time required to scan the entire mask field and the accuracy of distortion detection, an array of 7 × 7 sampling points is sufficient for the model resolution. Herein, the edge of the field points is included because the distortion is related to the whole field. At each field point, scanning data for distortion measurement are obtained 25 times to filter out the noise and coarse errors for a time period of 7 min. The wavelength used for distortion measurement in the experiments is 365 nm, and the numerical apertures (NA) were 0.58. An environmental control system is also

installed to keep the temperature at 300 K and the humidity at 56%. The Shack–Hartmann sensor has a focal length of 5.15 mm and a CCD radius of 3.2 mm.

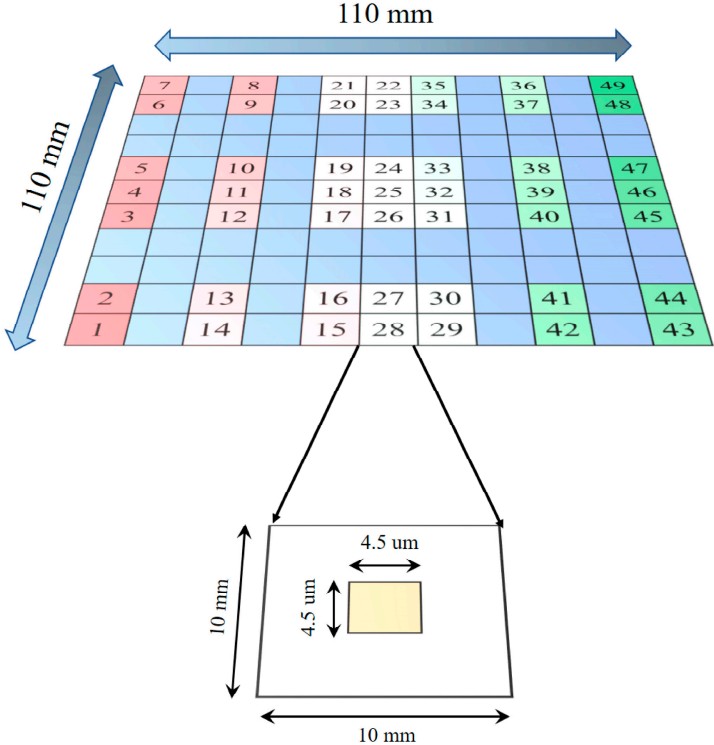

**Figure 5.** Mask design and sampling strategy.

Details of some experimental parameters are shown in Table 2; the whole mask field size is 110 mm × 110 mm. The effective field of the wafer is 22 mm × 22 mm by 5 × demagnification of the projection lens, in which 7 × 7 positions are sampled by dynamic scanning of the full field. The CCD camera model is a Basler aca1920-40GM, and the pixel size is 5.86 μm. The center point is defined as the origin or zero point, and the sampling interval is unequally spaced. The number of sampled field points for the validation stage prior to the experiment can be reduced to 5 × 5, but we must include the edge field points to effectively reduce the time cost and improve the efficiency. The experiment also proves that it has little impact on the calculation results, and we can effectively save 50% of the sampling time.

**Table 2.** The main specifications of the experiment.

| Parameter | | Specification |
| --- | --- | --- |
| Wavelength | | 365 nm |
| Illuminator | | LED |
| Numerical aperture (NA) | | 0.58 |
| Field size (wafer) | | 22 mm × 22 mm |
| Lens magnification | | 5× |
| Temperature | Global temperature stability | 300 ± 0.1 K@8 h |
| | Local temperature stability | 300 ± 0.01 K@4 h |
| Humidity | | 56% |
| CCD pixel size | | 5.86 μm |

## 5. Results and Discussion

The vector diagram intuitively indicates the quality of the data. For example, the distribution of the vector diagram can determine whether a specific type of error contributes in a relatively large proportion to the overall field error. This is extremely important for us to analyze the components of the error source in the experiment, so we can make certain adjustments to achieve reasonably good results. The purpose of our proposed model to detect these parameters is to adjust the objective distance, projection movable lens, and mechanical mounting position to reduce the distortion error. As shown in Figure 6a, the distortion vector map before adjustment can be also depicted. The distortion errors caused by translation, rotation, magnification, decentering distortion, thin prism distortion, and third-order radial distortion can be corrected gradually for the entire system. Figure 6b gives the distortion vector map after adjustment.

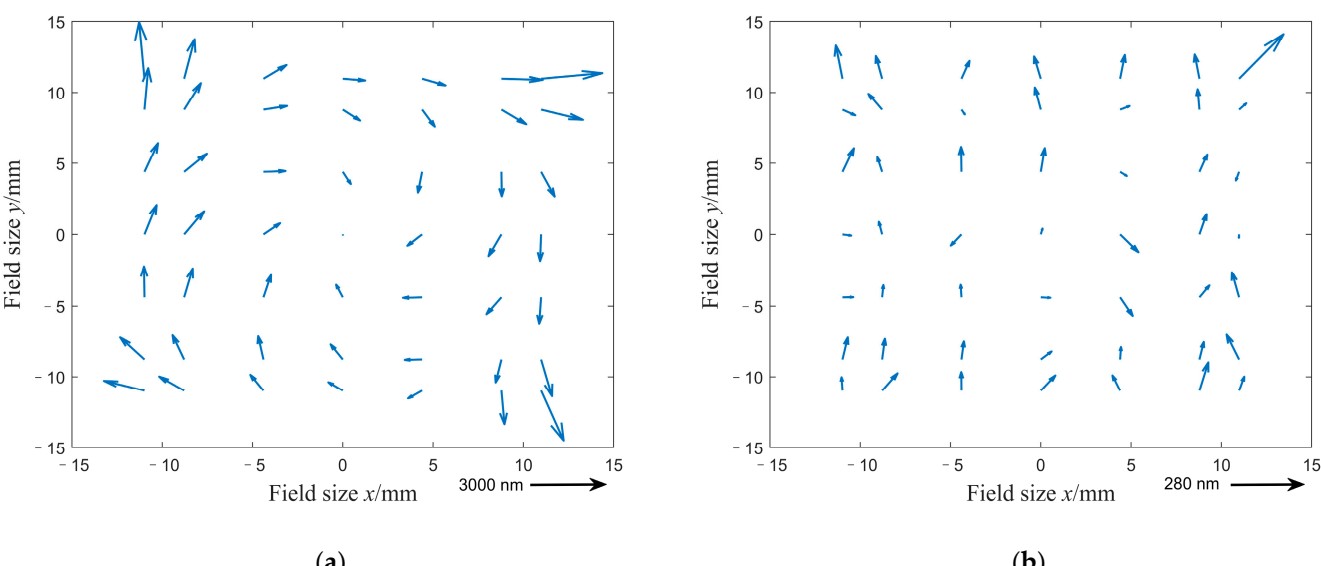

(**a**)            (**b**)

**Figure 6.** Pre- and post-adjustment distortion vector map. (**a**) The distortion vector map before the adjustment with the maximum distortion of 2634.61 nm and 3005.19 nm along the *x*-axis and *y*-axis, respectively. (**b**) The distortion vector map after the adjustment with the maximum distortion of 177.12 nm and 218.85 nm along the *x*-axis and *y*-axis, respectively.

As a result, the overall distortion offsets were measured by our Hartmann-based scheme according to Equations (10) and (11). Then, by resolving the distortion model as indicated in Equation (9), all distortion components of translation, rotation, magnification, decentering distortion, thin prism distortion, third-order radial distortion, distortion, and their repeatability (standard deviation) were obtained before and after adjusting the objective distance, projection movable lens *z*-distance, *x*–*z* angle, *y*–*z* angle, and mechanical installation inclination depicted in Table 3. The repeatability of the distortion error due to the detected parameters not only indicates the image quality of the objective lens but also guides the distortion calibration. For example, for a large rotation error, the angle between the mask and the scanning direction can be adjusted. For large thin prism distortion, the objective lens tilt angle for correction can be adjusted.

The short-term (2 h) and long-term (72 h) reproducibility for the 7 × 7 field points are calculated by this computational model and depicted in Figure 7. The reproducibility vector is scaled in the figure to denote the number of standard deviations of distortions correspondingly, with the maximum value of 142 nm for the short-term reproducibility and 143 nm for the long-term reproducibility, indicating that the system is quite stable.

**Table 3.** Repeatability of distortion data and distortion measurement before and after adjustment.

| Parameter | Unit | Before Adjustment | | After Adjustment | | Repeatability ($\sigma$) | |
|---|---|---|---|---|---|---|---|
| | | $x$ | $y$ | $x$ | $y$ | $x$ | $y$ |
| $T$ | nm | 3.35 | $-23.03$ | 10.27 | 5.21 | 12.70 | 28.75 |
| $M$ | nm/cm | $-35.44$ | $-34.62$ | 1.23 | 0.47 | 1.4 | 2.95 |
| $\theta$ | μrad | $-69.82$ | $-133.18$ | 0.15 | 0.21 | 2.06 | 2.70 |
| $p_{11}$ | nm/cm$^2$ | 30.04 | 37.56 | 18.45 | 23.67 | 3.19 | 3.83 |
| $p_{12}$ | nm/cm$^2$ | 27.77 | 96.87 | 0.89 | 0.73 | 2.46 | 4.21 |
| $q$ | nm/cm$^2$ | 106.15 | 55.90 | 1.15 | 0.43 | 4.45 | 4.99 |
| $D_3$ | nm/cm$^3$ | 1229.48 | 1218.80 | 1.26 | 0.62 | 4.97 | 7.44 |
| $d$ | nm | 2634.61 | 3005.19 | 177.12 | 218.85 | 27.63 | 51.08 |

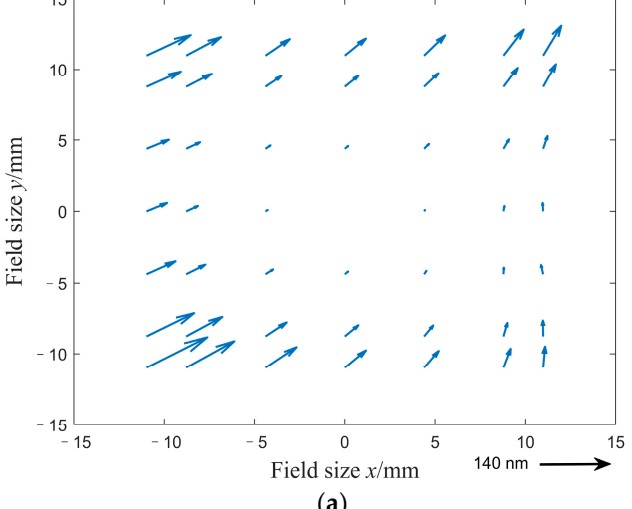

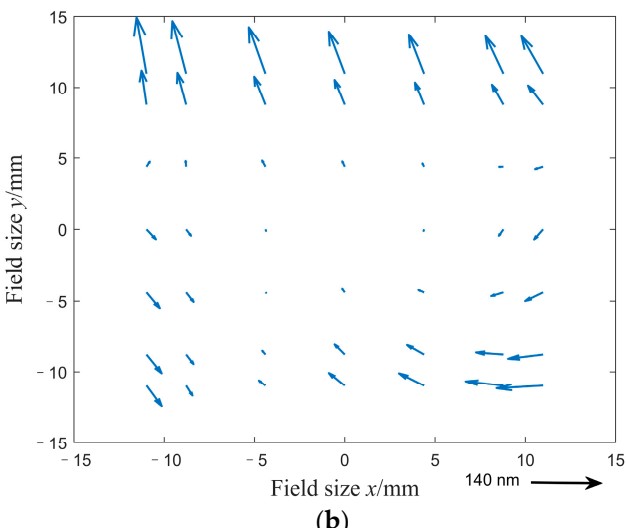

**Figure 7.** Short- (2 h) and long-term (72 h) reproducibility of distortion measurements. (**a**) The short-term reproducibility with the maximum value of 142 nm (2 h). (**b**) The long-term reproducibility with the maximum value of 143 nm (72 h).

The repeatability accuracy of the interferometer for global coordinate calibrations, as indicated in Equation (12), i.e., the degree of consistency between successive measurements for the same field point under the same conditions, is estimated in terms of standard deviation. The measuring system of the interferometer will have angle error, which is mainly affected by the machine assembly and environmental variables. Figure 8a shows the angle error of 49 field points in the three-dimensional direction, which is close to 2 urad. Figure 8b gives the repeatability accuracy of interferometer calibrations for all $7 \times 7$ sampling points along the $x$-direction and $y$-direction, where the maximum deviation of 34 nm is found at the 39th field point.

For the whole system, there are many factors that affect the distortion error. Figure 9 shows a fishbone diagram of the factors that affect the distortion error. Some factors originate from the subsystems of the illuminator, mask, projection lens, environment, stage, measurement tool, and metrology. Such factors might be coupled to affect the offset between the actual image point and the ideal image point. Among them, some factors are systematic errors, such as abbe error, cosine error, tilt error, rotation error, etc., which can be corrected and adjusted. Some component errors, such as uniformity error and flatness, cannot be corrected, because these errors are self-contained by the components themselves.

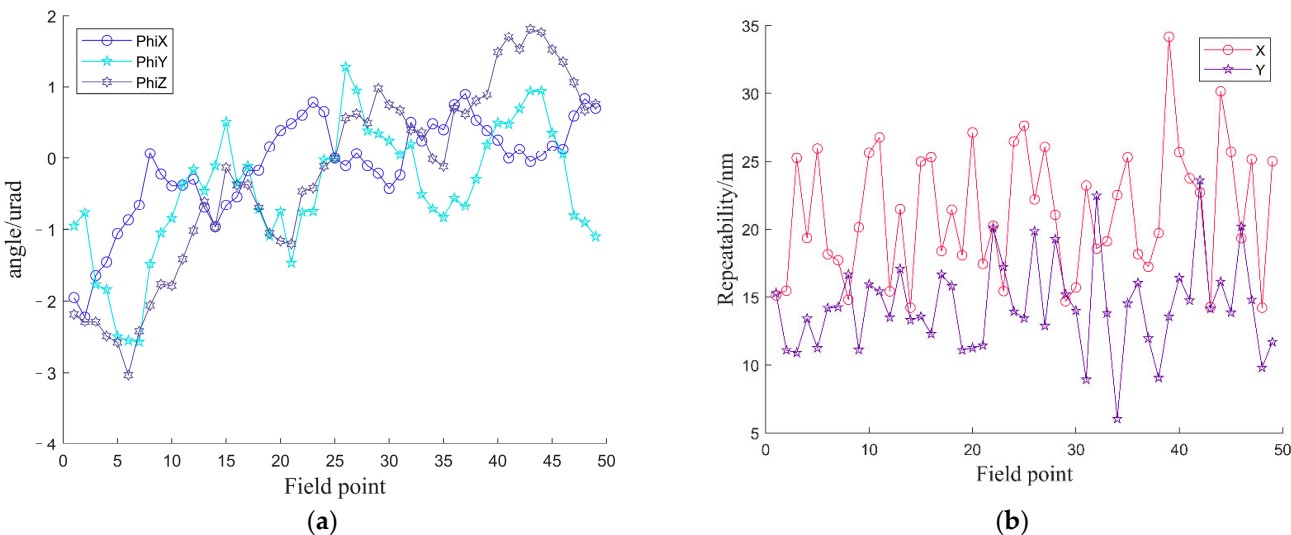

**Figure 8.** Angle error and measurement repeatability of interferometer. (**a**) Angle error of interferometer in three dimensions of global field of view. (**b**) The repeatability accuracy of the global measurements of coordinates for all $7 \times 7$ sampling points by the interferometer.

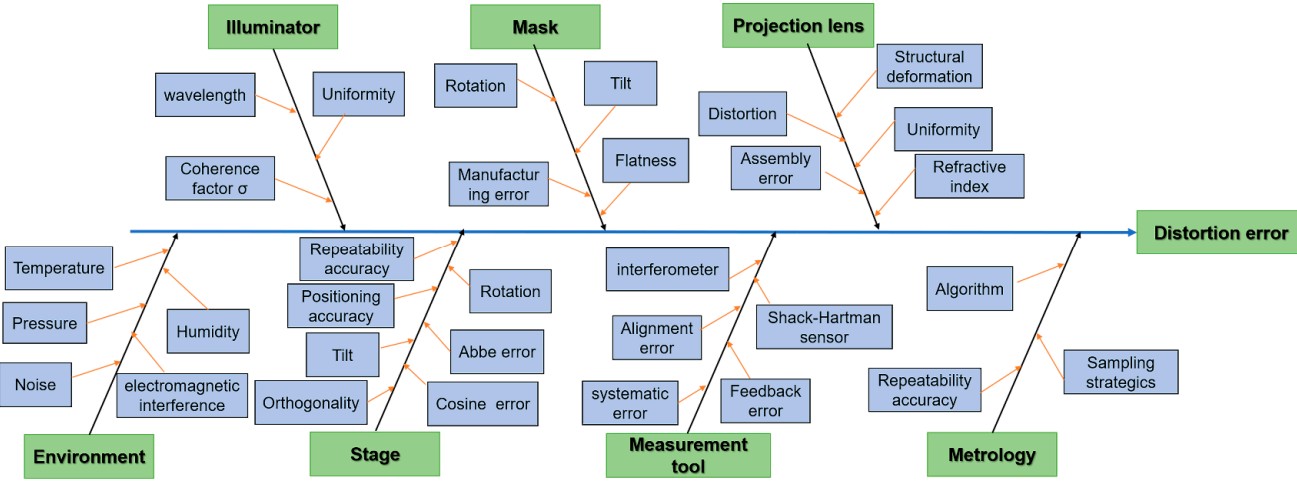

**Figure 9.** Fishbone map of distortion error.

In short, our scheme of distortion detection with the integration of a Shack–Hartmann sensor can detect all the decomposed components of distortion errors, including translation, rotation, magnification, decentering distortion, thin prism distortion, and third-order radial distortion. The distortion of $7 \times 7$ field points in the ideal position of the image plane and the actual position are scanned by the stage bearing the Shack–Hartmann sensor; the distortion detection model can give the distortion coefficients of the lens, and we adjust the distortion of the experimental lens' *x*-axis and *y*-axis from 2634.61 nm and 3005.19 nm to 177.12 nm and 218.85 nm through dynamic mirror adjustment. The repeatability accuracy of distortion detection was measured to be 51 nm, with the short-term and long-term reproducibility of 142 nm and 143 nm.

## 6. Conclusions

This study presents a projection lens distortion detection scheme based on Shack–Hartmann wavefront measurement, from which translation, rotation, magnification, decentering distortion, thin prism distortion, and third-order radial distortion can be detected. The system to be detected is a 5× projection lens with NA of 0.58 illuminated by an LED source with a wavelength of 365 nm. The actual image point formed in the image plane is

detected by the Shack–Hartmann sensor under the conditions of an ambient temperature of 300 K and humidity of 56%. The coordinates of the actual image point are recorded by the interferometer to detect the global position of the scanning stage for the $7 \times 7$ field points. Finally, the system of overdetermined equations is established and the parameters are resolved by using the least squares method. The experimental results reveal that the repeatability accuracy of distortion detection is 51 nm and the short-term and long-term reproducibility are 142 nm and 143 nm, respectively, with the distortion error of 177.12 nm and 218.85 nm along the *x*-axis and *y*-axis. This method is highly flexible for distortion measurement with portable and compactly integrated sensors, enabling the real-time and cost-efficient measurement of wave aberration and distortion.

Finally, the accuracy of the distortion detection scheme can be further improved by increasing the number of field points or improving the performance of experimental systems such as the stage (positioning, scanning, etc.), measurement tool (interferometer, sensors, etc.), and environmental control (temperature, humidity, pressure, etc.).

**Author Contributions:** Conceptualization, T.L. and H.Q.; writing—original draft preparation and experimental work, T.L.; writing—review and editing, S.Z. and J.W.; supervision and experimental work, L.C., J.L., J.D., X.Z., J.W. and S.H.; funding acquisition, J.W.; project administration, J.W. All authors have read and agreed to the published version of the manuscript.

**Funding:** This research was funded by the National Natural Science Foundation of China (NSFC, No: 61875201), the Science Project of Sichuan Province (No: 2022YFQ0011), the Youth Innovation Promotion Association of the Chinese Academy of Sciences (No: 2021380), and the project of the Western Light of Chinese Academy of Science for financial support.

**Institutional Review Board Statement:** Not applicable.

**Informed Consent Statement:** Not applicable.

**Data Availability Statement:** Data underlying the results presented in this paper are not publicly available at this time but may be obtained from the authors upon reasonable request.

**Acknowledgments:** The authors would like to thank Jinsheng Yang for the helpful discussions.

**Conflicts of Interest:** The authors declare no conflict of interest.

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
