# Peer review of "Distortion Detection of Lithographic Projection Lenses Based on Wavefront Measurement"

_photonics, doi:10.3390/photonics10020168_

Round 1
Reviewer 1 Report
Please see attached file for edits. Authors did a good job of writing a clear, easy to understand paper. However, references need a lot of work. Most references are missing - making it impossible to check if the authors are citing papers appropriately.

Author Response
Thank you for giving me the opportunity to submit revised draft of my manuscript titled “Distortion detection of lithographic projection lenses based on wavefront measurement, photonics-2055070.” We appreciate the time and effort that you have dedicated to providing your valuable feedback on our manuscript.
We have responded to your valuable comments and made corrections to the manuscript in the appropriate places.

Author Response

(The authors gave the same response as above.)

Reviewer 3 Report
The section's model is impossible to follow, the isn't a proper validation and presentation of the model, for example, the figure 1 is placed 6 times, among many other details that do not allow to read and even less to understand this paper.
Author Response

(The authors gave the same response as above.)

Round 2
Reviewer 3 Report
I think is necessary do a revision for improve the final version.
